# Pol µ dGTP mismatch insertion opposite T coupled with ligation reveals promutagenic DNA repair intermediate

Melike Çağlayan[1,2] & Samuel H. Wilson[1]

Incorporation of mismatched nucleotides during DNA replication or repair leads to transition or transversion mutations and is considered as a predominant source of base substitution mutagenesis in cancer cells. Watson-Crick like dG:dT base pairing is considered to be an important source of genome instability. Here we show that DNA polymerase (pol) µ insertion of 7,8-dihydro-8′-oxo-dGTP (8-oxodGTP) or deoxyguanosine triphosphate (dGTP) into a model double-strand break DNA repair substrate with template base T results in efficient ligation by DNA ligase. These results indicate that pol µ-mediated dGTP mismatch insertion opposite template base T coupled with ligation could be a feature of mutation prone non-homologous end joining during double-strand break repair.

[1] Genome Integrity and Structural Biology Laboratory, National Institutes of Health, National Institute of Environmental Health Sciences, Research Triangle Park, NC 27709, USA. [2] Present address: Department of Biochemistry and Molecular Biology, University of Florida, Gainesville, FL 32610, USA. Correspondence and requests for materials should be addressed to M.Ç. (email: caglayanm@ufl.edu) or to S.H.W. (email: wilson5@niehs.nih.gov)

Watson-Crick like tautomeric and anionic dG:dT mismatches have important roles in spontaneous replication errors[1–4]. Deoxyguanosine triphosphate (dGTP) in the nucleotide pool can be oxidized by reactive oxygen and nitrogen species (ROS) leading to 8-oxo-7,8-dihydro-2′-deoxyguanosine (8-oxodGTP)[5]. Incorporation of 8-oxodGTP into genomic DNA during DNA replication or repair can lead to genome instability[6]. This is associated with deleterious effects including mutagenesis in cells, and has implications in various diseases such as cancer[7]. It has been suggested that DNA repair intermediates can be transferred directly from one step to the next during DNA repair[8]. This hand off mechanism was envisioned as a means to prevent accumulation of toxic repair intermediates in cells[9]. Previous work with DNA polymerase (pol) β had revealed that 8-oxodGMP insertion opposite template bases C or A resulted in a frayed structure containing 3′-end 8-oxoG at the polymerase active site[10]. This impairs base excision repair (BER) and leads to ligation failure and 5′-adenylation (i.e., an addition of AMP to the 5′-end of the repair intermediate)[11]. We also previously reported DNA strand break formation and cell death in pol β mouse embryonic fibroblast cells under oxidative stress[12]. Double-strand breaks (DSBs) are among the most dangerous forms of DNA damage; they lead to genome instability, cell lethality, and carcinogenesis[13]. The repair of oxidative stress-induced DSBs through nonhomologous end joining (NHEJ) involves DNA synthesis by pol μ, pol λ, or pol β, and DNA ligation by DNA ligase I or X-ray cross complementing protein 4 (XRCC4)/DNA ligase IV complex[14–16].

In the present study, we show that DNA ligase efficiently ligates the NHEJ repair intermediate following insertion of Watson-Crick like mismatches dGTP or 8-oxodGTP opposite template base T by pol μ in vitro. We also show that ligation failure is observed with the ligation of 3′-preinserted mismatches 8-oxodG or dG opposite template base T by DNA ligase alone. The results point to a facilitated hand off of the Watson-Crick like dG:dT mismatch repair product in the presence of pol μ during DSB repair.

## Results

**Pol μ dG:dT mismatch insertion coupled with DNA ligation.** In this study, we investigated the impact of two Watson-Crick like promutagenic mismatches on the downstream steps during DNA repair. For this purpose, we used a coupled repair assay to evaluate the effect of pol μ 8-oxodGTP or normal nucleotide dGTP incorporation into a single-nucleotide gapped DNA substrate with template base T on the ligation step in vitro. This system mimics the channeling of a model end joining repair intermediate from pol μ to DNA ligase during DSB repair (Supplementary Fig. 1, Supplementary Table 1). We found that pol μ dGMP or 8-oxodGMP insertion opposite template base T was abundant and enabled ligation without 5′-adenylation (Fig. 1a). In both cases, the insertion and ligation products appeared at the earliest time point (10 s), and the repair intermediate was ligated efficiently (Fig. 1b, Supplementary Fig. 2). There was no significant difference in the amounts of these products with DNA ligase I or DNA ligase IV/XRCC4 complex (Supplementary Fig. 3), indicating that pol μ was able to accommodate the dG:dT and 8-oxodG:dT base pairs in a fashion that enabled both DNA synthesis and ligase activities. In a control experiment with correct dATP insertion opposite template base T, there was efficient conversion of inserted product to ligated product (Supplementary Fig. 4a, b). However, we observed reduced ligation and formation of abortive ligation product after pol μ 8-oxodGMP insertion opposite template bases C or A (Supplementary Fig. 5a). The abortive ligation products were more abundant with template base A than

template base C (Supplementary Fig. 5b, c). Overall, these results indicated that pol μ was able to stabilize the 8-oxodGMP insertion product opposite template base T, allowing DNA ligase to function more efficiently than with 8-oxodGMP insertions opposite bases C or A (Supplementary Fig. 6).

**Comparison of 8-oxodGTP:dT insertion coupled with ligation.** In contrast to this picture with pol μ, we observed ligation failure after 8-oxodGMP insertion opposite template base T by the other X-family repair DNA polymerases pol β and pol λ (Supplementary Fig. 7). This indicates that these pols were not able to accommodate the wobble or Watson-Crick like dG:dT mismatch in a fashion that enables DNA ligase action, as seen with pol μ. Control experiments with DNA ligase alone revealed that both adenylation and ligation require insertion of a nucleotide at the 3′-edge in the nicked DNA substrate (Supplementary Fig. 4c, d). We previously demonstrated that 8-oxodGMP insertion by pol β or pol λ opposite bases C or A confounds DNA ligation during BER[11].

**Effect of 3′-preinserted dG:dT mismatch on DNA ligation.** Finally, we were interested in examining the repair of nicked DNA substrates with 3′-preinserted mismatches 8-oxodG or dG opposite template base T in the ligation reactions including DNA ligase alone. This system enables us to observe how DNA ligase handles the nicked DNA substrate with 3′-dG:dT base pairs that mimics a model end joining repair intermediate after pol μ mismatch insertions (Supplementary Fig. 8, Supplementary Table 1). The results indicated that DNA ligase shows differences in ligation of the repair intermediate, depending on the nature of damaged and mispaired ends (Fig. 2a, Supplementary Fig. 9). Interestingly, DNA ligase I was able to ligate the 3′-end with dG more efficiently than with 3′-8oxodG (Fig. 2b). We observed similar results with DNA ligase IV/XRCC4 complex (Supplementary Fig. 10). In a control experiment with nicked substrate including 3′-preinserted correct dA:dT base pair, only ligation products were obtained (Supplementary Fig. 11).

## Discussion

The stable dG:dT base pair is well known, and this mismatch is considered to be mutation prone in cells[3,4]. Multiple alternate base pairing options can stabilize the G:T combination, and it is evident that the pol μ active site is able to accommodate this mismatched base pair for insertion. Since 8-oxodGTP and dGTP are able to serve as substrate in a similar fashion, it is likely the strain imposed by the 8-oxo group is well accommodated by pol μ. The difference in our findings between a coupled pol μ-DNA ligase system (Fig. 1) vs ligation reactions (Fig. 2) revealed that pol μ dGTP mismatch insertion opposite T in the template could facilitate hand off the repair intermediate to DNA ligation step for efficient NHEJ. The impact of the ligation step in NHEJ fidelity including the significance of DNA ligase IV in sensing diverse strand breaks and guiding them for ligation after ionizing radiation has been reported[17–19]. Our results here demonstrate that pol μ dG:dT mismatch insertion coupled with ligation could be a promutagenic event leading to genomic instability during double-strand DNA repair.

## Methods

**Enzyme purification.** Recombinant human DNA polymerases were overexpressed in *Escherichia coli* and purified as previously described for pols β[20], λ[21] and μ[22], respectively. Briefly, for pol β and pol μ purification, cells were lysed by sonication in a lysis buffer including 25 mM Tris(HCl), pH 8.0, 500 mM NaCl, 5% glycerol and 1 mM DTT. The lysate was clarified by centrifugation, and the polymerases were purified by chromatography on a glutathione-Sepharose 4B column at 4 °C. The polymerases were eluted by TEV cleavage overnight at 4 °C

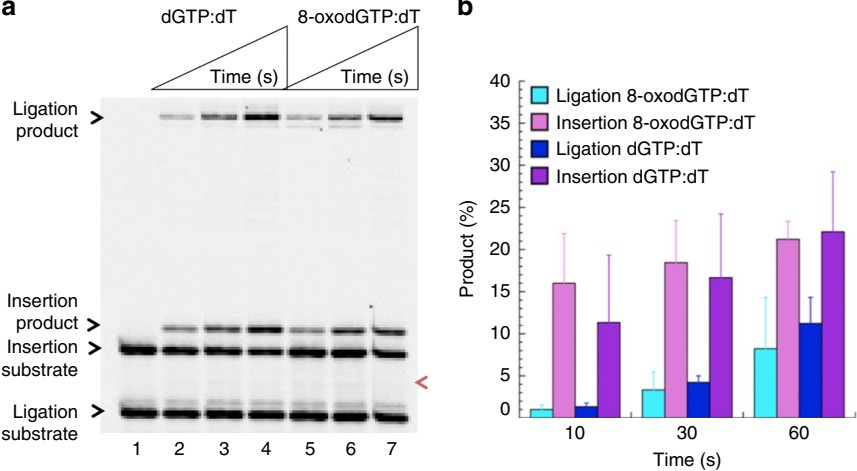

**Fig. 1** Pol μ dGMP and 8-oxodGMP insertion coupled with ligation. **a** Lane 1 is the minus enzyme control for the single-nucleotide gapped DNA substrate with template base T. Lanes 2–4 and 5–7 are the reaction products in the presence of dGTP and 8-oxodGTP, respectively, and correspond to time points of 10, 30, and 60 s. The position of 5′-adenylate product is indicated as magenta arrow. **b** Graph shows time-dependent changes in the products of insertion (pink and purple for 8-oxodGTP and dGTP insertions, respectively) and ligation (cyan and blue for ligation following insertion of 8-oxodGTP and dGTP, respectively). The data represent the average of four independent experiments ± SD. Corresponding uncropped gel image and presentation of the data in line graph format are shown in Supplementary Fig. 2

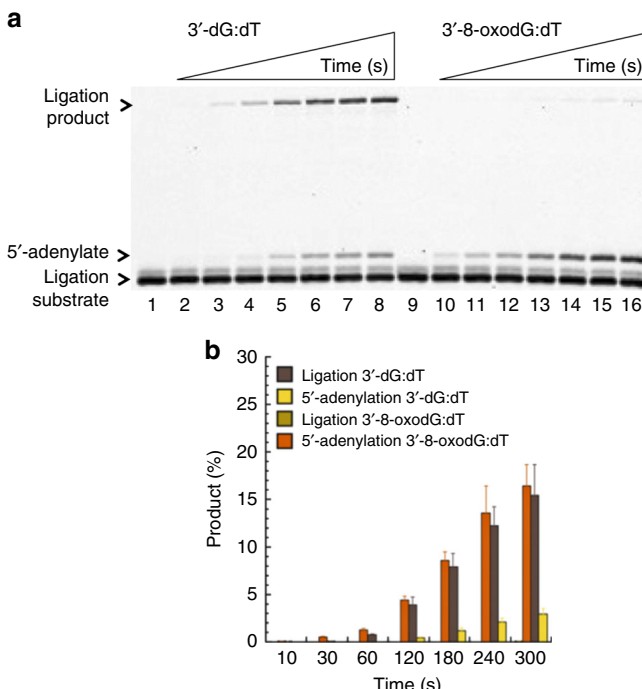

**Fig. 2** Ligation of 3′-preinserted dG and 8-oxodG. **a** Lanes 1 and 9 are the minus enzyme controls for the nicked DNA substrate with 3′-dG and 3′-8-oxodG, respectively. Lanes 2–8 and 10–16 are the reaction products, and correspond to time points of 10, 30, 60, 120, 180, 240, and 300 s. **b** Graph shows time-dependent changes in the products of ligation (light and dark brown for ligation of 3′-8-oxodG and 3′-dG, respectively) and 5′-adenylation (orange and yellow for 5′-adenylation following ligation of 3′-8-oxodG and 3′-dG, respectively). The data represent the average of four independent experiments ±SD. Corresponding uncropped gel image is shown Supplementary Fig. 9

and then purified by chromatography on Superdex 200 in a buffer containing 25 mM Tris(HCl), 100 mM NaCl, 5% glycerol, 1 mM DTT and 1 mM EDTA. The fractions containing the respective enzyme were combined, concentrated using a centrifugal filter unit, and then exchanged into a storage buffer containing 25 mM

Tris(HCl), 100 mM NaCl, 5% glycerol and 1 mM DTT. The final enzyme samples were frozen in dry ice and stored in aliquots at –80 °C. For pol λ purification, the clarified lysate was first passed through a Q-Sepharose column, and then the enzyme was purified using a nickel column at 4 °C. The enzyme was eluted with an imidazole gradient, and the purified enzyme was polished using Mono S column chromatography with NaCl gradient elution at 4 °C. The fractions containing pol λ were combined, and the enzyme was concentrated and stored as described above.

**A coupled DNA polymerase-ligase assay**. The single-nucleotide gapped DNA substrates with template bases A, C, or T (Supplementary Table 1) were prepared as described[12]. Briefly, the 5′-end FAM-labeled upstream oligonucleotide (19-mer) was annealed with the 3′-end FAM-labeled downstream oligonucleotide (17-mer) in the presence of the template oligonucleotide (37-mer) as represented in Supplementary Fig. 1. The coupled assays including both DNA polymerase (pol μ, pol β, or pol λ) and DNA ligase (DNA ligase I or DNA ligase IV/XRCC4 complex) were performed under steady-state conditions as described[12]. Briefly, the reaction mixture (10 μl in final volume) included 50 mM Tris(HCl), pH 7.5, 100 mM KCl, 10 mM MgCl$_2$, 1 mM ATP, 100 μg ml$^{-1}$ BSA, 10% glycerol, 1 mM DTT, and 200 μM of 8-oxodGTP (Jena Bioscience), dGTP or dATP (NEB), 300 nM DNA substrate, and 100 nM enzyme mixture. The reaction was initiated by addition of both enzymes that had been preincubated at 37 °C for 3 min. The reaction mixture was incubated at 37 °C for the indicated time points and then mixed with an equal volume of 95% formamide dye. The reaction products were seperated, and the data were analyzed as described[12].

**Ligation assay**. The nicked DNA substrates with template base T (Supplementary Table 1) were prepared as described[12]. Briefly, the upstream oligonucleotide with 3′-8-oxodG, 3′-dG, or 3′-dA (18-mer) was annealed with the 3′-end FAM-labeled downstream oligonucleotide (16-mer) in the presence of the template oligonucleotide (34-mer) as represented in Supplementary Fig. 8. The ligation assays including DNA ligase (DNA ligase I or DNA ligase IV/XRCC4 complex) alone were performed under steady-state conditions as described[12]. Briefly, the reaction mixture (10 μl in final volume) included 50 mM Tris(HCl), pH 7.5, 100 mM KCl, 10 mM MgCl$_2$, 1 mM ATP, 100 μg ml$^{-1}$ BSA, 10% glycerol, 1 mM DTT, 300 nM DNA substrate, and 50 nM DNA ligase. The reaction was initiated by addition of DNA ligase. The reaction mixture was incubated at 37 °C for the indicated time points, and then mixed with an equal volume of 95% formamide dye. The reaction products were seperated, and the data were analyzed as described[12].

## Data availability

All relevant data are available from the authors upon reasonable request.

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

## Acknowledgements

We thank Rajendra Prasad and David Shock for the purified proteins DNA polymerases λ and μ, respectively. DNA ligase IV/XRCC4 complex was a generous gift from Alan E. Tomkinson (University of New Mexico). The authors thank Bill Beard for his comments on the manuscript. This work was supported by the Intramural Research Program of the NIH, National Institute of Environmental Health Sciences (Project numbers Z01 ES050158 and Z01 ES050159).

## Author contributions

M.Ç. designed the project, performed the experiments, and wrote the manuscript. S.H.W. commented on the results and revised the manuscript.

## Additional information

**Competing interests:** The authors declare no competing interests.

