## [Peer Review File · Nature Communications]

Reviewers' comments:

Reviewer #1 (Remarks to the Author):

The title of this article might imply that the article would be of interest to only a specialized readership. However, the work is of broad interest and relevance, as it reveals an important source of genomic instability due to the action of a member of the X-family of DNA polymerases (which lack proofreading domains) in relation to the processing of double-strand breaks by non-homologous end joining. This processing requires closely-coupled "hand off" to a DNA ligase, as had been shown in Wilson's laboratory for the action of pol beta in base excision-repair. Here it is shown that DNA polymerase mu can mis-incorporate either dGTP or the oxidized precursor, 8-oxo-dGTP to dT in the template, followed by efficient ligation in a model DNA substrate. In contrast, they observed poorer and abortive ligation after insertion of 8-oxo-dGTP opposite templates, dC or dA. Interestingly, there was ligation failure after insertion opposite dT by pol beta or pol lambda. The stable dG:dT mismatch is known to be mutation prone, and this study shows that it can be stabilized by pol mu.

The paper is generally well-written and the material in the Supplements is appropriately and accurately discussed in the text. I have attached a version of the text, that includes my suggestions for minor revision. These include what could be a clarification of the title, in which "Pol mu dG:dT misinsertion ..." does not distinguish which nucleotide is inserted, unless there is a convention that the first designation, dG, must be the one that is inserted opposite the second one.

Reviewer #2 (Remarks to the Author):

The manuscript "Polmu dG:gT misinsertion coupled with ligation reveals promutagenic DNA intermediate during double strand repair" addresses the ability of polmu to incorporate oxidized dGTP (8-oxodGTP) into gapped oligonucleotides. The authors find that in contrast to other polymerases (addressed in a previous NatComm paper) polmu incorporates 8-oxodGTP equally well as dGTP and dATP across a T which suggests that polmu could cause promutagenic mismatches at such NHEJ intermediates.

The data in this manuscript is of high quality. All appropriate controls have been executed and the most relevant variables have been tested. Kinetics show that Polmu does not appear to discriminate much between the correct dA, the dG or oxodG and ligation is not affected either (within the errors), overall a very distinctive and interesting finding.

This reviewer's suggestion is to strengthen the link or specificity to double strand break repair and genome instability. The authors assessed activities/kinetics with the help of single nucleotide gapped or nicked oligonucleotides as substrates. These are possible intermediates in NHEJ DSB repair and polmu has been reported to be involved in NHEJ justifying the model. Yet, it would be helpful if the authors could explain this better for the readership (page 2 line 38 to 41 - in particular as the title might prompt a different expectation.

- Schemes: helpful to indicate the difference in length between the substrates for first time readers of such assays. Also helpful to illustrate the investigated step / intermediate in double strand break repair in the Sup-Figures so to place this study into context while limiting text.
- Most graphs show kinetics, which means the data (Y axis) points depend on each other (even though "sampled" by different lanes in the assays). This should be presented in line graphs. This also eases the interpretations of the results.
- Dotted reference lines that indicate the incorporation / ligation kinetics of the dGTP or 8-oxodGTP:dT on the other Figures (e.g. Sup-Fig 3) help to compare the kinetics across the different

conditions (and figures) that have been shown and thereby highlight how similar they actually are.

- Even though not too important with these particular results/conclusions, formally one should specify error bars (SEM, SD, range ?) and describe the statistics that has been used.
- Methods do not describe how the proteins were prepared. Would it be possible to show data confirming that this is pol mu?

July 27, 2018

Re: NCOMMS-18-13035-T

The title of original submission "Pol μ dG:dT misinsertion coupled with ligation reveals promutagenic DNA intermediate during double strand repair"

The revised title of the manuscript "Pol μ dGTP mismatch insertion opposite T coupled with ligation reveals promutagenic DNA"

Point-by-point responses to the Referees' comments are as follows:

Referee: 1

Comments for the Authors

The title of this article might imply that the article would be of interest to only a specialized readership. However, the work is of broad interest and relevance, as it reveals an important source of genomic instability due to the action of a member of the X-family of DNA polymerases (which lack proofreading domains) in relation to the processing of double-strand breaks by non-homologous end joining. This processing requires closely-coupled "hand off" to a DNA ligase, as had been shown in Wilson's laboratory for the action of pol beta in base excision-repair.

Here it is shown that DNA polymerase mu can mis-incorporate either dGTP or the oxidized precursor, 8-oxo-dGTP to dT in the template, followed by efficient ligation in a model DNA substrate. In contrast, they observed poorer and abortive ligation after insertion of 8-oxo-dGTP opposite templates, dC or dA. Interestingly, there was ligation failure after insertion opposite dT by pol beta or pol lambda. The stable dG:dT mismatch is known to be mutation prone, and this study shows that it can be stabilized by pol mu. The paper is generally well-written and the material in the Supplements is appropriately and accurately discussed in the text.

Response:

We thank the Referee for these positive comments. We appreciate the recognition that ligation after dGTP mismatch insertion opposite T could be stabilized by pol μ leading to mutation prone processing of double-strand breaks during non-homologous end joining. We have revised the title and manuscript to accommodate the Referee's comments, as summarized below:

Point 1) I have attached a version of the text that includes my suggestions for minor revision. These include what could be a clarification of the title, in which “Pol mu dG:dT misinsertion ...” does not distinguish which nucleotide is inserted, unless there is a convention that the first designation, dG, must be the one that is inserted opposite the second one.

Response:

We accepted all comments/suggestions by the Referee in the main text, and the text is now revised accordingly. Accordingly, in the main text of the revised manuscript, we changed the designation dA, dC, and dT to A, C, and T when we describe DNA polymerase nucleotide incorporation opposite template bases A, C, or T. Also, we revised the title of the manuscript as suggested: Pol μ dGTP mismatch insertion opposite T coupled with ligation reveals promutagenic DNA intermediate.

We appreciate the assistance of the Referee very much.

Referee: 2

Comments for the Authors

The manuscript “Pol mu dG:dT misinsertion coupled with ligation reveals promutagenic DNA intermediate during double strand repair” addresses the ability of pol mu to incorporate oxidized dGTP (8-oxodGTP) into gapped oligonucleotides. The authors find that in contrast to other polymerases (addressed in a previous Nat Comm paper) pol mu incorporates 8-oxodGTP equally well as dGTP and dATP across a T which suggests that pol mu could cause promutagenic mismatches at such NHEJ intermediates.

The data in this manuscript is of high quality. All appropriate controls have been executed and the most relevant variables have been tested. Kinetics show that Pol mu does not appear to discriminate much between the correct dA, the dG or oxodG and ligation is not affected either (within the errors), overall a very distinctive and interesting finding.

Response:

We thank the Referee for these positive comments. We appreciate the recognition of the high quality of the data in the study. We have revised the manuscript to accommodate the Reviewer's comments, as summarized below.

Point 1) This reviewer's suggestion is to strengthen the link or specificity to double strand break repair and genome instability. The authors assessed activities/kinetics with the help of single nucleotide gapped or nicked oligonucleotides as substrates. These are possible intermediates in NHEJ DSB repair and pol μ has been reported to be involved in NHEJ justifying the model. Yet, it would be helpful if the authors could explain this better for the readership (page 2 line 38 to 41 - in particular as the title might prompt a different expectation.

Response: In order to address the Referee's point, we have revised a portion of the manuscript (lines 38-41 in the original submission of the manuscript) to better explain the DNA polymerase-ligase coupled assay system used in the study. We have revised the text accordingly (lines 40-45 in the revised manuscript). We also revised a portion of the manuscript to better explain the ligase assay system (lines 76-78 in the revised manuscript). We appreciate the Referee's assistance in clarifying these points. Also, we revised the title of the manuscript as follows: Pol μ dGTP mismatch insertion opposite T coupled with ligation reveals promutagenic DNA intermediate.

Point 2) Schemes: helpful to indicate the difference in length between the substrates for first time readers of such assays. Also helpful to illustrate the investigated step / intermediate in double strand break repair in the Sup-Figures so to place this study into context while limiting text.

Response: In order to address the Referee's point, we have revised the Scheme. We prepared two schemes and illustrated the DNA polymerase coupled ligation assay (Supplementary Scheme 1) and the ligation assay (Supplementary Scheme 2) separately. In both schemes, we indicated the differences between the lengths of the DNA substrates that mimic double strand break repair intermediates before and after pol μ dGTP insertion opposite template base T, and difference in the nicked DNA intermediate on which DNA ligase joins the ends with the 3'-dG:dT mismatch.

Point 3) Most graphs show kinetics, which means the data (Y axis) points depend on each other (even though “sampled” by different lanes in the assays). This should be presented in line graphs. This also eases the interpretations of the results.

Response: In general, we agree with the Referee on this point about use of line graphs. However, we believe that for the current study the quantification data can be presented and understood more clearly as bar graphs, instead of line graphs. However, in order to address the Referee’s point regarding representing the data in line graphs, we re-plotted the data in Figure 1 and now present this in a new Figure, as Supplementary Figure 1, in the revised supplementary data. Because of this addition, Supplementary Figure numbers are changed in the main text of the revised manuscript.

Point 4) Dotted reference lines that indicate the incorporation / ligation kinetics of the dGTP or 8-oxoGTP:dT on the other Figures (e.g. Sup-Fig 3) help to compare the kinetics across the different conditions (and figures) that have been shown and thereby highlight how similar they actually are.

Response: Supplementary Figure 3 in the original submission of the manuscript shows the incorporation and ligation after pol μ 8-oxodGTP insertions opposite template base C and A. We presented the main data (the ligation after pol μ 8-oxodGTP and dGTP insertion opposite template base T) in Figure 1. However, in order to address the Referee’s point regarding the comparison of the data on the same plot, we prepared a figure (Supplementary Figure 5 in the revised supplementary data) that highlights how similar or different the incorporation and ligation reactions are after pol μ 8-oxodGTP insertion opposite template base T vs. A for the time points of 10, 30, 60 s. However, we were not able to compare the data for pol μ 8-oxodGTP insertion opposite template base C as we obtained the reaction products at later time points (0.5, 1, 2, 3, 4, and 5 min).

Point 5) Even though not too important with these particular results/conclusions, formally one should specify error bars (SEM, SD, range ?) and describe the statistics that has been used.

Response: We appreciate the Referee’s assistance very much. The error bars show the SD of the data represented in the figures. The Referee’s point has been addressed by describing the statistics that have been used in the study. Accordingly, we have added this information (*i.e.*, the data represent the average of four independent experiments \pm SD) into the figure legends in the revised manuscript.

Point 6) Methods do not describe how the proteins were prepared. Would it be possible to show data confirming that this is pol mu?

Response: The X-family DNA polymerases used in the study are DNA pol β , pol λ , and pol μ . We previously published a protocol paper that describes the step-by-step procedure for the coupled repair assay with purified proteins pol β and DNA ligase I (Caglayan and Wilson, 2017). We used the same assay system in this current study. The purified proteins pol β and pol λ used in this study were prepared in our group as described previously (Beard and Wilson, 1995; Braithwaite E.K. *et al.*, 2010). We have added the references below to the reference list of the revised manuscript (refs 1, 2 and 6 in the revised supplementary methods).

- Caglayan M. and Wilson S.H. (2017) *In vitro* assay to measure DNA polymerase β nucleotide insertion coupled with the DNA ligation during base excision repair. *Bio Protocol* 7, pii: e2341.
- Beard W.A. and Wilson S.H. (1995) Purification and domain-mapping of mammalian DNA polymerase beta. *Methods in Enzymology* 262: 98–107.
- Braithwaite E.K., Kedar P.S., Stumpo D.J., Bertocci B., Freedman J.H., Samson L.D. and Wilson, S.H. (2010) DNA Polymerases β and λ mediate overlapping and independent roles in base excision repair in mouse embryonic fibroblasts. *Plos One* 5(8): e12229

As we indicated in the acknowledgement section of the revised manuscript, the purified protein pol μ was a gift. We also added the reference (ref 3 in the revised Supplementary Methods) below that describes the purification and characterization of the protein.

- Moon A.F., Pryor J.M., Ramsden D.A., Kunkel T.A., Bebenek K., and Pedersen L.C. (2014) Sustained active site rigidity during synthesis by human DNA polymerase μ . *Nat. Struc. Mol. Biol.* 21(3): 253-260.